# Dulaglutide 1.5 mg Significantly Improves Glycemic Control and Lowers LDL-Cholesterol and Body Weight in Romanian Patients with Type 2 Diabetes

**DOI:** 10.3390/jcm14103536

**Published:** 2025-05-18

**Authors:** Amelian Madalin Bobu, Serban Turliuc, Andrei Ionut Cucu, Alina Onofriescu, Cristina Gena Dascalu, Claudia Florida Costea, Emilia Patrascanu, Anca Petruta Morosan, Anca Haisan, Carmen Nicoleta Filip, Roxana Covali, Catalin Mihai Buzduga, Gina Botnariu, Irina Iuliana Costache Enache

**Affiliations:** 1Department of Cardiology, Clinical Emergency Hospital St. Spiridon, 700111 Iasi, Romania; amelian.bobu@gmail.com (A.M.B.); irina.costache@umfiasi.ro (I.I.C.E.); 2Faculty of Medicine, University of Medicine and Pharmacy Grigore T. Popa Iasi, 700115 Iasi, Romania; alina.onofriescu@umfiasi.ro (A.O.); cdascalu_info@yahoo.com (C.G.D.); claudia.costea@umfiasi.ro (C.F.C.); patrascanu.emilia@umfiasi.ro (E.P.); petruta2001@yahoo.com (A.P.M.); anca.haisan@umfiasi.ro (A.H.); carmen.filip@umfiasi.ro (C.N.F.); catalinbuzduga@gmail.com (C.M.B.); ginabotnariu66@gmail.com (G.B.); 3Faculty of Medicine and Biological Sciences, University Stefan cel Mare of Suceava, 720229 Suceava, Romania; andrei.cucu@usm.ro; 4Emergency Clinical Hospital Prof. Dr. Nicolae Oblu, 700309 Iasi, Romania; 5Department of Diabetes and Metabolic Diseases, Clinical Emergency Hospital St. Spiridon, 700111 Iasi, Romania; 6Emergency Department, Clinical Emergency Hospital St. Spiridon, 700111 Iasi, Romania; 7Faculty of Medical Bioengineering, University of Medicine and Pharmacy Grigore T. Popa Iasi, 700115 Iasi, Romania; ana.covali@umfiasi.ro; 8Department of Endocrinology, Clinical Emergency Hospital St. Spiridon, 700111 Iasi, Romania

**Keywords:** dulaglutide, glucagon-like peptide-1 receptor agonist, incretin mimetic, type 2 diabetes, glycemic control, body weight control, LDL-cholesterol control

## Abstract

**Background:** Dulaglutide is a glucagon-like peptide-1 (GLP-1) receptor agonist administered subcutaneously once a week, developed through recombinant DNA technology, and prescribed as an add-on to diet and exercise for managing type 2 diabetes mellitus in adults. In several clinical trials, once-weekly dulaglutide has demonstrated reductions in cardiovascular risk associated with diabetes, as well as improvements in glycemic control and weight reduction. The scope of this study was to evaluate the effect of dulaglutide 1.5 mg on glycemic control, weight management, and LDL-cholesterol levels in patients with uncontrolled type 2 diabetes mellitus. **Methods:** We retrospectively reviewed the medical records of 55 patients with inadequately controlled type 2 diabetes mellitus who were on oral antidiabetic agents and insulin, and who were additionally treated with dulaglutide 1.5 mg. We monitored fasting plasma glucose and glycated hemoglobin (HbA1c) at baseline and at 6, 12, and 24 months after initiating dulaglutide treatment. Weight, body mass index, and LDL-cholesterol were assessed at baseline and after 24 months of dulaglutide therapy. **Results:** Treatment with dulaglutide resulted in significant improvements in fasting plasma glucose and HbA1c after 6 months (*p* < 0.001), 12 months (*p* < 0.001), and 24 months (*p* < 0.001). A significant weight reduction was observed after 24 months of dulaglutide therapy (−3.3 kg; *p* < 0.001). In addition, we observed a significant reduction in LDL-cholesterol after 24 months (*p* < 0.001). **Conclusions:** Our data demonstrate that dulaglutide 1.5 mg significantly improves glycemic control, reduces body weight, and lowers LDL-cholesterol in Romanian patients with inadequately controlled type 2 diabetes.

## 1. Introduction

Type 2 diabetes mellitus (T2DM) is among the most prevalent chronic diseases worldwide, affecting approximately 387 million people in 2014, and it is estimated that this number will reach 592 million by 2035 [1]. The condition is managed using a variety of multimodal approaches, including dietary modifications and physical activity, with a primary emphasis on achieving and maintaining glycemic control [2,3,4].

Incretins, including glucagon-like peptide-1 (GLP-1) and glucose-dependent insulinotropic polypeptide (GIP), are hormones produced in the gut that are released following nutrient ingestion [5]. GLP-1 is a 30-amino acid peptide hormone synthesized by endocrine cells in the intestinal epithelium and, upon release, it promotes insulin secretion—a phenomenon known as the incretin effect [5]. Incretin-based therapies, such as injectable GLP-1 receptor agonists (GLP-1 RAs) and oral dipeptidyl peptidase-4 (DPP-4) inhibitors, are being increasingly utilized in the management of T2DM [6].

Dulaglutide, a recombinant human GLP-1 RA administered once weekly, was approved in 2014 for the treatment of T2DM as an adjunct to exercise and diet [7]. Studies have shown that dulaglutide significantly reduces glycated hemoglobin (HbA1c), fasting plasma glucose (FPG), and postprandial plasma glucose levels in patients with T2DM [8,9,10,11,12]. In several large phase III clinical trials, dulaglutide, both as monotherapy and as adjunctive therapy, has been shown to improve glycemic control in adults with T2DM [13,14,15,16,17,18,19,20].

Once-weekly dulaglutide administration has been linked to significant (*p* < 0.05), dose-dependent improvements in beta-cell function, as measured by HOMA-%β, compared to placebo after 12 or 16 weeks of treatment in patients with T2DM [13,15]. Furthermore, increases from baseline in HOMA-%β were significantly greater (*p* < 0.001) in patients treated with dulaglutide 0.75 mg or 1.5 mg once weekly compared to those receiving metformin monotherapy at both 26 and 52 weeks [13]. When administered in combination with metformin, with or without pioglitazone, dulaglutide resulted in significant increases in HOMA-%β (*p* < 0.001) [16]. In other studies, dulaglutide resulted in significantly greater increases in HOMA-%β (*p* < 0.001) compared to daily sitagliptin at 52 weeks [14] and 104 weeks [21]. Moreover, peripheral insulin sensitivity, assessed using the homeostasis model assessment for insulin sensitivity (HOMA2-%S), was significantly reduced (*p* ≤ 0.01) in patients receiving dulaglutide 0.75 mg and 1.5 mg once weekly compared to those treated with metformin [13]. This finding suggests that the primary mechanism by which dulaglutide reduces blood glucose levels is through the improvement of pancreatic beta-cell function [13].

Other studies have demonstrated that dulaglutide administration has beneficial effects on cardiovascular risk factors. Specifically, dulaglutide has been shown to reduce body weight [8,9,10,11,12], lower blood pressure [22], and improve certain cardiovascular risk biomarkers in patients with T2DM, including total cholesterol and LDL-cholesterol [14,16,22], as well as triglycerides [22].

Moreover, recent studies have highlighted the cardiovascular protective effects of dulaglutide in patients with T2DM, showing a reduction in the risk of major adverse cardiovascular events and improvements in vascular function [23,24,25,26].

Furthermore, in psychiatry, interest in dulaglutide and other GLP-1 RAs has recently increased due to their beneficial effects on body weight, especially in patients treated with antipsychotics, who are at higher risk for obesity and metabolic syndrome [27,28,29,30,31,32].

In Romania, the first study to demonstrate the beneficial effects of dulaglutide in the Romanian population was conducted by Ruda et al. in 2023 [33]. In this observational study, the authors analyzed the effects of dulaglutide in patients with T2DM and, after 12 months of treatment, observed a significant reduction in HbA1c by 2%, a decrease in body weight by 3.5 kg, and a reduction in insulin dosage among patients [33].

The scope of this study was to evaluate the effect of once-weekly dulaglutide 1.5 mg on glycemic control, body weight, and LDL-cholesterol levels in Romanian patients with T2DM inadequately controlled with antihyperglycemic medication.

## 2. Material and Methods

### 2.1. Study Design and Patients

We conducted a retrospective review of medical records from 55 patients with T2DM, inadequately controlled with antihyperglycemic therapy (oral antidiabetic agents and insulin), who initiated treatment with dulaglutide 1.5 mg. The patients were selected from an Outpatient Diabetes Center serving T2DM patients from the eastern region of Romania (Bacău, Romania). All patients were followed up at 6, 12, and 24 months after the initiation of dulaglutide treatment, during the period from 1 January 2019 to 31 December 2024. Comorbid conditions (e.g., obesity and hypertension) and complications associated with T2DM (peripheral diabetic polyneuropathy, diabetic retinopathy, and diabetic nephropathy) were identified in accordance with the diagnostic guidelines of the American Diabetes Association [34]. The exclusion criteria for patients in our study included individuals with type 1 diabetes, those under 18 years of age, patients with comorbidities other than diabetic and cardiovascular conditions, and those who experienced adverse reactions after initiating treatment with dulaglutide that led to discontinuation of therapy.

The diagnosis of diabetic peripheral polyneuropathy was made clinically, based on patient-reported symptoms and physical examination findings, which assessed (1) deep tendon reflexes (Achilles and patellar), and (2) vibratory, thermal, painful, fine touch, and proprioceptive sensation. The tools used included the 10 g monofilament, a 128 Hz tuning fork, pinprick testing, and a basic thermal test.

The diagnosis of diabetic neuropathy was established according to the criteria of the American Diabetes Association, requiring at least one symptom or sign of neuropathy and the presence of a symmetric, distal, and progressive pattern of nerve involvement. The diagnosis of diabetic retinopathy was established by an ophthalmologist based on fundus examination using an ophthalmoscope. The diagnostic criteria used were those of the Early Treatment Diabetic Retinopathy Study (ETDRS), focusing on the presence of microaneurysms, retinal hemorrhages, hard and soft exudates, abnormal vascular proliferations, and diabetic macular edema.

The diagnosis of diabetic nephropathy was based on the presence of albuminuria in two out of three urine tests performed more than three months apart, along with the assessment of renal function using serum creatinine and estimated glomerular filtration rate (eGFR).

The study protocol was approved by the local Ethics Committee of the Outpatient Diabetes Center, in compliance with the World Medical Association’s Declaration of Helsinki and institutional regulations (Approval code No. 080/09.12.2024). Informed consent from patients was not required, as the study was retrospective in nature.

### 2.2. Study Protocol and Assessments

We monitored the following variables before and after initiating dulaglutide treatment: age, sex, duration of T2DM, comorbidities and diabetes-related complications at the time of diagnosis, and the type of treatment (non-insulin antidiabetic drugs, insulin). In addition, height and weight were measured for each patient, and body mass index (BMI) was calculated accordingly. BMI classification was performed according to the World Health Organization guidelines [35]. No patients in our study were classified as underweight. Other laboratory data monitored included FPG, HbA1c, and LDL-cholesterol. At the time of dulaglutide therapy initiation, all patients were receiving HMG-CoA reductase inhibitor therapy. FPG was measured using blood-glucose meters by the diabetologist during the patient’s clinic visit, while HbA1c and LDL-cholesterol were assessed using commercially available methods. FPG and HbA1c were monitored at 6, 12, and 24 months after once-weekly dulaglutide 1.5 mg was added to the patients’ previous antihyperglycemic treatment regimen. Weight, BMI, and LDL-cholesterol were measured both before initiating dulaglutide treatment and after 24 months of therapy.

### 2.3. Statistical Analysis

Statistical analysis was performed using the Statistical Package for the Social Sciences (SPSS) version 24.0 for Windows (SPSS Inc., Chicago, IL, USA). Descriptive statistics were computed for numerical variables, while frequency distributions were determined for categorical variables. Data collected at baseline, 6, 12, and 24 months were compared using the paired-samples *t*-test and the Wilcoxon signed-rank test for related samples. Additional statistical analyses included Friedman’s two-way analysis of variance for related samples, independent-samples *t*-test (Student’s *t*-test), the Mann–Whitney U test, one-way ANOVA, and the Kruskal–Wallis test. A *p*-value ≤ 0.05 was considered statistically significant.

## 3. Results

### 3.1. Baseline Characteristics

Of the 55 patients included in the study, 54.5% (n = 30) were male, and 58.2% (n = 32) were from urban areas. The median age was 58.2 years (SD: 10.1; range 33–77 years), and the median duration of T2DM was 9.5 years (range 3–19 years). The average age for men was 56.8 years (CI 95%: 53.56, 60.18), while for women it was 59.8 years (95% CI: 55.16 to 64.52). Male patients had a longer average duration of diabetes compared to female patients (10.2 vs. 8.6 years). The confidence interval values were 95% for both men (95% CI: 8.36 to 12.04) and women (95% CI: 6.90 to 10.46).

When analyzing the evolution of biological parameters by sex, we observed that male patients showed, on average, greater reductions in BMI, body weight, and glycemic parameters (FPG and HbA1c). However, Student’s *t*-test and the Mann–Whitney test showed that none of these differences were statistically significant (all *p*-values > 0.05). Thus, dulaglutide treatment had a comparable effect between sexes, with no significant differences in the evolution of BMI, body weight, FPG, HbA1c, or LDL-cholesterol (Figure 1).

Of the 55 patients, 52.7% (n = 29) had essential hypertension. Regarding complications, the most common was diabetic polyneuropathy (27.3%, n = 15), followed by diabetic retinopathy (16.4%, n = 9) and diabetic nephropathy (n = 10). In the entire cohort, 25.5% (n = 14) of patients had no diabetes-related complications.

At the time of dulaglutide initiation, 23.6% (n = 13) of patients were treated with metformin alone, 61.9% (n = 34) were treated with metformin and sulfonylureas, and 9.1% were on metformin and DPP-4 inhibitors. Out of the 55 patients, only 7 (12.7%) were receiving insulin as part of their antihyperglycemic regimen. All these treatment schemes had positive effects on weight, BMI, HbA1c, FPG, and LDL-cholesterol; however, the differences between treatment groups were not statistically significant. The characteristics of the patient cohort are presented in Table 1.

### 3.2. Glycemic and LDL-Cholesterol Control

At the initiation of dulaglutide treatment, the mean FPG was 179 mg/dL (range 112 mg/dL–280 mg/dL, SD: 36.96), and after 24 months, the mean FPG decreased to 136.4 mg/dL (range 90 mg/dL–203 mg/dL, SD: 19.56). Thus, the mean FPG decreased by −18 mg/dL at 6 months (SD: 28.71, *p* < 0.001), −30 mg/dL at 12 months (SD: 23.76, *p* < 0.001), and −43 mg/dL at 24 months (95% CI: −25.46 to −18.19) (*p* < 0.001) from the initiation of dulaglutide treatment (Figure 2).

At the initiation of treatment, the mean HbA1c was 8.43% (range 6.5–11%, SD: 1.15), decreasing after 6 months to 7.78% (range 6.2–10.5%, SD: 0.89), after 12 months to 7.28% (range 6–10.4%, SD: 0.93), and after 24 months to 6.82% (range 5.48–10.1%, SD: 0.89; 95% CI: −21.24 to −14.94). Thus, HbA1c significantly decreased by −0.67% at 6 months (*p* < 0.001), by −1.15% at 12 months (*p* < 0.001), and by −1.61% at 24 months (*p* < 0.001) from the initiation of treatment with dulaglutide (Figure 3).

At the initiation of dulaglutide treatment, the mean LDL-cholesterol level was 112.92 mg/dL (44.6 mg/dL–343.6 mg/dL, SD: 46.01), and after 24 months, it had decreased to 99.22 mg/dL (50.0 mg/dL–221.0 mg/dL, SD: 30.38, 95% CI: −13.15 to −3.32) (*p* < 0.001) (Table 2).

### 3.3. Weight Control

At the initiation of treatment, the mean weight of the patients in the group was 95.2 kg (range 64–167 kg, SD: 18.17) and after 24 months of treatment with dulaglutide, it was 91.8 kg (range 58–134 kg, SD: 16.02; (95% CI: −4.56 to −1.94). The mean weight loss was 3.3 kg (Table 2).

Regarding BMI at the initiation of dulaglutide treatment, the mean value was 33.48 kg/m^2^ (range 24.2–53 kg/m^2^, SD: 6.19), IQR = [28.6, 37.7], and after 24 months of treatment, the mean BMI was 32.28 kg/m^2^ (range 22.37–55 kg/m^2^, SD: 5.89), IQR = [27.72, 35.79]. Analyzing the effects of dulaglutide on weight control, we found a statistically significant reduction after 24 months, with a mean decrease in BMI of 1.2 kg/m^2^ (*p* < 0.001) (Table 2).

Table 3 presents the paired comparisons of key clinical parameters measured at baseline and during follow-up periods. Prior to selecting the appropriate statistical tests, we assessed the distribution of continuous variables using the Shapiro–Wilk test. For variables that followed a normal distribution—specifically, BMI and HbA1c at baseline and after 6 months—we used the paired samples *t*-test. For variables that did not meet the normality assumption, including body weight, FPG, HbA1c at later time points, and LDL-cholesterol, we applied the non-parametric Wilcoxon signed-rank test. This approach allowed us to more accurately reflect central tendency and significance within the constraints of the data.

As shown, statistically significant improvements were observed in all parameters across the follow-up periods. BMI and weight significantly decreased after two years of treatment (*p* < 0.001), while HbA1c showed marked reductions as early as 6 months, with further improvements sustained through 2 years (*p* < 0.001 at all intervals). Similarly, FPG levels declined consistently at each time point, and LDL-cholesterol was significantly reduced at 2 years (*p* < 0.001). These findings indicate both short- and long-term metabolic benefits of treatment. The statistical methods used are clearly labeled in the table with the corresponding test statistic and *p*-values. (Table 3).

The results show a significant improvement in all analyzed clinical parameters, with negative mean differences reflecting a beneficial long-term effect of the intervention. All 95% confidence intervals were entirely below zero, indicating that the observed changes are not due to chance but are statistically significant. For example, HbA1c decreased by a mean of –18.09 mmol/mol (95% CI: –21.24 to –14.94), and FPG by –21.83 mg/dL (95% CI: –25.46 to –18.20), supporting the presence of improved and sustained glycemic control over the two-year period. Weight and LDL-cholesterol reductions, although more modest, followed the same favorable trend.

The mean differences between baseline and 2-year values for each parameter, along with the 95% confidence intervals, are shown in Figure 4.

## 4. Discussion

Our observational study assessed the effects of dulaglutide 1.5 mg on glycemic control, weight management, and LDL-cholesterol levels in Romanian patients with T2DM inadequately controlled with antihyperglycemic therapy. We identified a statistically significant improvement in glycemic control (FPG and HbA1c) at 6, 12, and 24 months after initiating treatment with dulaglutide, a reduction in weight at 24 months, and a significant decrease in LDL-cholesterol levels at 24 months.

Regarding the duration of diabetes, in our study group, the mean duration was 9.5 years (range 3–19 years), a longer duration compared to the patients enrolled in the AWARD-3 study, where the mean duration was 2.6 years [13], and also longer than in the Japanese trials, where the mean duration of diabetes was 6.6 years [20] and 8.8 years [19].

### 4.1. The Effect of Dulaglutide on Glycemic Control

Recent studies, particularly the 11 AWARD clinical trials (Assessment of Weekly Administration of Dulaglutide), have provided evidence supporting the use of dulaglutide in reducing HbA1c levels in patients with T2DM, with the continuation of these trials over time highlighting the ongoing interest in optimizing dulaglutide treatment in patients with T2DM [36,37]. Another significant study is the REWIND trial, which monitored cardiovascular events in 9.091 participants across 24 countries, who were randomized to receive either dulaglutide or a placebo [38]. In this study, HbA1c decreased by −0.61% in patients managed with dulaglutide over a median follow-up period of 5.4 years. In a literature review, Robinson et al. reported an HbA1c reduction of 0.5–2.2% from baseline over 3 to 24 months of dulaglutide treatment across 20 studies [39], results similar to ours, as we observed a mean reduction in HbA1c of −1.1% at 12 months and −1.6% at 24 months.

In 2023, Ruda et al. conducted the first study in Romania evaluating the impact of dulaglutide treatment in Romanian patients with T2DM, but over a 12-month period [33]. Comparing their results with ours, we observed that in our study, HbA1c decreased more slowly, with reductions of −0.6% and −1.15% at 6 and 12 months, respectively, compared to their study, where the authors reported greater reductions in HbA1c of −1.3% and −2% at 6 and 12 months after initiating dulaglutide treatment [33]. Similar differences in results were also identified regarding FPS. Thus, at 6 and 12 months, we observed mean reductions in FPG of −18 mg/dL and −30 mg/dL, respectively, whereas Ruda et al. reported reductions of −56 mg/dL and −55.5 mg/dL at the same time intervals [33]. One possible explanation could be the difference in treatment regimens between the patient groups in the two studies. In our cohort, only 12.7% (n = 7) of patients were on insulin therapy, compared to the other study, where 70% (n = 35) of patients were receiving insulin therapy [33].

Some authors have reported significant reductions in HbA1c levels as early as 6 months after initiating treatment with dulaglutide [40,41]. In line with these reported data, our results also demonstrated the beneficial effects of dulaglutide treatment on glycemic control in patients with T2DM, with statistically significant values observed as early as the first 6 months after treatment initiation, showing a mean reduction in HbA1c of −0.6%.

Among the patients in our study, optimal glycemic control (HbA1c < 7%) was attained by 19.3% at 6 months, 38.4% at 12 months, and 58% at 24 months, indicating a progressive improvement over time. By comparison, patients in the AWARD clinical trials achieved HbA1c levels < 7% at significantly higher rates. For instance, in the AWARD-1 study, 78% of patients reached HbA1c < 7% at 6 months [16], 61% in AWARD-5 [21], 68% in AWARD-6 [36], and 55.3% in AWARD-8 [42]. Similarly, at 12 months after initiating dulaglutide, 38.47% of our patients achieved HbA1c < 7%, a much lower percentage compared to AWARD-1, which reported 57% [16], AWARD-5 (58%) [14], AWARD-6 (53.2%) [15], and AWARD-11 (58.6%) [36]. These significant differences could be explained by geographic variations across regions of Europe. In this regard, the EURODIAB study, which included 3.250 patients from 16 countries across Europe, including patients from Romania, concluded that HbA1c levels were higher in centers from Eastern Europe compared to those in Southern or Northwestern Europe. Moreover, patients from Eastern Europe (n = 539) had the highest levels of HbA1c, LDL-cholesterol, and triglycerides compared to Europeans from Southern or Northwestern Europe. As for BMI, the average was similar across all regions of Europe [43]. Furthermore, another study that included 1.853 patients from Eastern European countries such as Slovenia, Croatia, Serbia, Bulgaria, and Romania observed that the level of glycemic control in this region was poor, reaching the target HbA1c levels, with more than 60% of patients failing to achieve optimal control of T2DM [44]. Therefore, improving prevention and managing the complications of T2DM is essential, especially in Eastern European regions, and in this regard, further studies to establish specific genetic backgrounds are needed. Additionally, two other recent studies conducted in Romania showed that the majority of Romanian patients with T2DM often fail to achieve glycemic control targets [45,46].

In Romania, patients’ adherence to nutritional recommendations remains low, and this negatively affects the effectiveness of antidiabetic treatment [47,48], leading to increased HbA1c levels, the key marker in diabetes monitoring. Numerous previous studies conducted on the Romanian population have reported a high prevalence of elevated HbA1c, indicating poor glycemic control [48,49]. This issue is exacerbated by various socio-economic and patient-related factors. Among these, the lack of communication between patients and healthcare providers contributes to poor adherence, as well as a lack of understanding of the disease and its management, which can result in poor self-monitoring and inadequate dietary control [47]. Furthermore, poor glycemic control among Romanian patients is also due to inadequate adherence to prescribed regimens and required lifestyle changes [48,49], with sedentary lifestyles and unhealthy dietary habits, which are prevalent in Romania, further complicating the management of T2DM [48].

### 4.2. The Impact of Dulaglutide on Lowering LDL-Cholesterol

Our results showed that the reduction in LDL-cholesterol was another positive sign of dulaglutide treatment, indicating an improvement in cardiovascular risk. To date, various studies in the literature have demonstrated a significant reduction in LDL-cholesterol in patients treated with dulaglutide compared to those receiving conventional therapy or a placebo [14,16,25]. Moreover, Tuttolomondo et al., in a study involving 56 patients monitored over a 9-month period, demonstrated that diabetic patients receiving conventional therapy combined with dulaglutide 1.5 mg showed more favorable outcomes compared to those treated with conventional therapy alone, and not only experienced favorable metabolic effects (such as reductions in body weight, BMI, FPS, and HbA1c) but also improvements in vascular health markers, including arterial stiffness (pulse wave velocity), augmentation index, and endothelial function (reactive hyperemia index). Thus, the authors concluded that dulaglutide could reduce the risk of stroke in patients with T2DM not only through lipid-lowering effects but also through direct vascular mechanisms [25].

In a recent 12-month study investigating the cardiometabolic effects of dulaglutide in 65 Caucasian patients with T2DM, Al Refaie et al. reported that treatment with GLP-1 RAs, including dulaglutide, led not only to a significant reduction in LDL-cholesterol and triglyceride levels but also to a significant increase in serum adiponectin levels, indicating reduced insulin resistance and inflammatory activity. Additionally, the authors observed an important decrease in carotid artery mean intima-media thickness, highlighting the importance of GLP-1 RAs in improving cardiovascular risk [50].

Regarding the combined effects of both statins and dulaglutide on lowering cholesterol, it is important to note that all 55 patients in our study group had been on maximum-dose statin therapy for at least one year, yet their LDL-cholesterol levels were still not within normal limits. The introduction of dulaglutide into the patients’ treatment regimens also led to a reduction in LDL-cholesterol, with several current clinical studies reporting the favorable effect of dulaglutide on the lipid profile [25,38,51]. In this context, isolating the effects of dulaglutide is challenging when both therapies are used concurrently, as statins can independently reduce LDL-cholesterol [52,53], and the results should therefore be interpreted with caution.

### 4.3. The Effects of Dulaglutide on Weight Control

Another desirable effect of dulaglutide treatment is weight loss, particularly in overweight or obese patients. In this regard, in our study, dulaglutide had a positive impact on weight control. The mean weight loss of patients after 24 months of follow-up was −3.3 kg. We observed a significant reduction in body weight after 24 months of dulaglutide treatment, which also led to a significant decrease in BMI. It is worth mentioning that the mean BMI in our study at the initiation of dulaglutide treatment was 33.48 kg/m^2^, consistent with all the AWARD studies, in which enrolled patients had a mean BMI ranging between 31–34 kg/m^2^ [14,15,16,17,18].

In the literature, several pragmatic and uncontrolled studies have reported weight loss outcomes similar to those observed in our patient group. For instance, the TROPHIES study reported a weight loss of −3.5 kg after 24 months of follow-up [54], and Kim et al., in a Korean study, reported a weight loss of −3.19 kg after a follow-up period of 33.1 months [55].

In the AWARD-11 study, patients receiving dulaglutide 1.5 mg experienced a weight loss of 3.5 kg after 12 months [37], a finding consistent with the results observed in our study. Across 15 observational studies monitoring patients treated with dulaglutide over periods ranging from 3 to 12 months, reported weight loss varied between 2.1 kg and 6.4 kg [39]. Furthermore, other authors have reported more rapid weight loss after 6 months of dulaglutide treatment, ranging between 2.05 and 2.9 kg [40,41,56]. Variations in weight loss outcomes between our study and others may be explained by differences in patients’ previous antidiabetic therapies at the time dulaglutide was initiated, along with possible socio-cultural influences affecting lifestyle habits, treatment adherence [57], and genetic differences between geographical regions of Europe.

Regarding weight loss during treatment with GLP-1 RAs, the literature has reported that they reduce both fat mass and lean body mass, although the extent of muscle mass loss varies across studies. A 2024 meta-analysis of 19 randomized controlled trials found that GLP-1 RAs reduced both fat mass and lean body mass, but with proportional muscle loss [58]. In this context, some authors argue that the reduction in muscle mass may be adaptive rather than maladaptive: the decrease in muscle volume is proportional to the degree of weight loss and the disease status [59]. On the other hand, various authors have demonstrated beneficial effects of GLP-1 RAs on skeletal muscle, both in preclinical and clinical studies, showing that GLP-1 RA treatment inhibits muscle atrophy and promotes muscle protein synthesis [60,61], helps maintain muscle mass [62,63,64,65], or even increases it in patients with T2DM [66].

The effects of dulaglutide on muscle mass are variable: some studies suggest muscle loss [67], while others do not identify a significant negative effect [68]. On the other hand, some preclinical studies have shown potential functional benefits of dulaglutide treatment. For example, the administration of dulaglutide and exendin-4 in mice with induced muscle atrophy led to increased muscle mass and function. The authors concluded that GLP-1 RAs alleviate muscle atrophy by suppressing myostatin and muscle atrophy-related factors, as well as by stimulating the expression of myogenic factors [69]. Although the authors highlight the potential usefulness of GLP-1 RA treatment in neuromuscular diseases associated with atrophy, further studies are still needed to clearly determine the sarcopenic effect of GLP-1 RAs.

While our study was not powered to perform detailed subgroup analyses, it is important to acknowledge a key nuance regarding body composition changes associated with GLP-1 RAs. Although these agents significantly reduce total body weight and fat mass, recent evidence indicates that their effect on relative lean mass is minimal. A recent network meta-analysis [70] found that approximately 25% of total weight loss with GLP-1 RAs and dual GLP-1 and Gastric Inhibitoy Polypeptide RA is due to a reduction in lean mass; however, the proportion of lean mass relative to total body weight remains stable. This suggests that while absolute lean mass decreases, overall body composition is largely preserved. Recognizing this distinction is critical for accurately interpreting the clinical impact of these therapies.

The most common adverse effects of dulaglutide reported in clinical trials are gastrointestinal in nature, including nausea, vomiting, diarrhea, abdominal pain, and constipation, with nausea being the most frequently cited reason for treatment discontinuation [13,15]. These reactions occur predominantly during the initiation period, are partially dose-dependent, and may be mitigated through gradual dose escalation. Hypersensitivity reactions, although rare, may require immediate discontinuation of therapy. These include skin rashes, pruritus, angioedema, and, in very rare cases, anaphylactic reactions. Other adverse effects, such as injection site reactions (pain, erythema, induration), are less frequent but can contribute to reduced adherence in patients reluctant to use injectable therapies [15]. A rare risk of pancreatitis has also been reported, warranting caution in patients with a history of such episodes [71].

In addition to objective clinical manifestations, subjective discomfort, fear of adverse effects, and negative perceptions of treatment may lead to voluntary therapy discontinuation, even in the absence of severe reactions [72]. In phase III clinical trials (the AWARD program), discontinuation rates for dulaglutide ranged from 5% to 10%, with gastrointestinal causes predominating [13,15]. In real-world practice, these rates may be higher, particularly in the absence of proper counseling and individualized dose titration plans. To reduce the risk of discontinuation, initiation with the minimum dose (0.75 mg), followed by gradual escalation, along with close monitoring during the first few weeks of therapy, is recommended. Patient education plays a crucial role in preventing treatment dropout.

In our study, patients who were unable to tolerate adverse effects and discontinued dulaglutide treatment were excluded. Most patients in the study group experienced nausea during the first weeks of dulaglutide initiation; however, the symptoms were mild, self-limiting, and did not lead to treatment discontinuation. Regarding treatment with other oral antidiabetic drugs and insulin, sulfonylureas and insulin in particular are well known for their potential to induce weight gain, especially when used as monotherapy or in more intensive combinations [73,74]. On the other hand, metformin—considered the first-line therapy in T2DM—has neutral or even mildly weight-reducing effects and is often continued alongside GLP-1 RAs [75]. DPP-4 inhibitors, although weight-neutral, may indirectly affect glycemic response if replaced with GLP-1 RAs [55]. In this context, post-hoc analyses of the AWARD studies (e.g., AWARD-2, -4, and -10) show significant differences in the efficacy of dulaglutide depending on concomitant medication. For example, its effectiveness in reducing HbA1c is greater when added to metformin compared to basal insulin [17]. Therefore, the evaluation of dulaglutide’s effects on weight and glycemia must be interpreted in the context of prior and concurrent antihyperglycemic therapies. We consider the most important limitations of our study to be the heterogeneity of oral antidiabetic treatments administered to the patients and the relatively small patient sample.

## 5. Conclusions

In conclusion, this retrospective analysis demonstrates that once-weekly administration of dulaglutide 1.5 mg yields significant and sustained improvements in glycemic control, body weight, and LDL-cholesterol levels in patients with inadequately controlled T2DM. These findings reinforce the therapeutic value of dulaglutide as part of a multifactorial management strategy for T2DM. Further prospective, controlled studies are warranted to elucidate the long-term impact of dulaglutide on microvascular and macrovascular diabetic complications, including its potential to slow the progression of retinopathy, nephropathy, and neuropathy. Thus, future research directions may include the identification of genetic or metabolic profiles associated with enhanced therapeutic response, and the investigation of dulaglutide’s effects on extended lipid profiles and inflammatory status, and consequently on cardiovascular risk, as well as its potential neuroprotective effects against diabetes-associated cognitive decline.

## Figures and Tables

**Figure 1 jcm-14-03536-f001:**
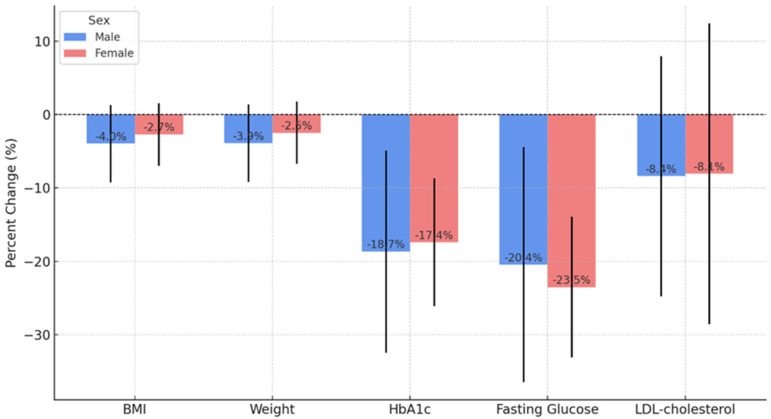
Comparison of percentage changes between gender for anthropometric and metabolic parameters after 24 months of dulaglutide treatment (n = 55).

**Figure 2 jcm-14-03536-f002:**
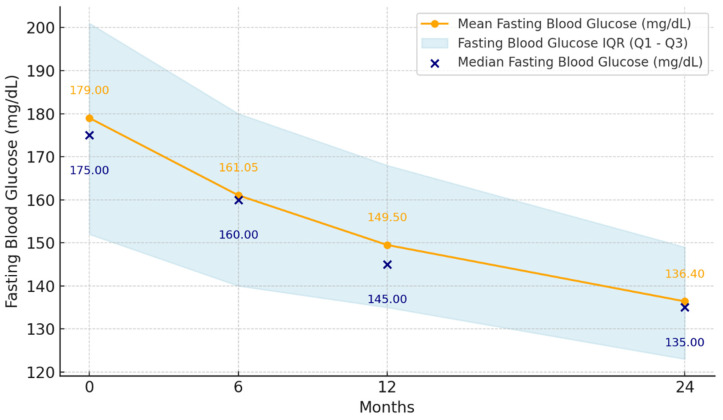
Mean fasting plasma glucose levels at 6, 12, and 24 months after initiation of dulaglutide treatment (n = 55).

**Figure 3 jcm-14-03536-f003:**
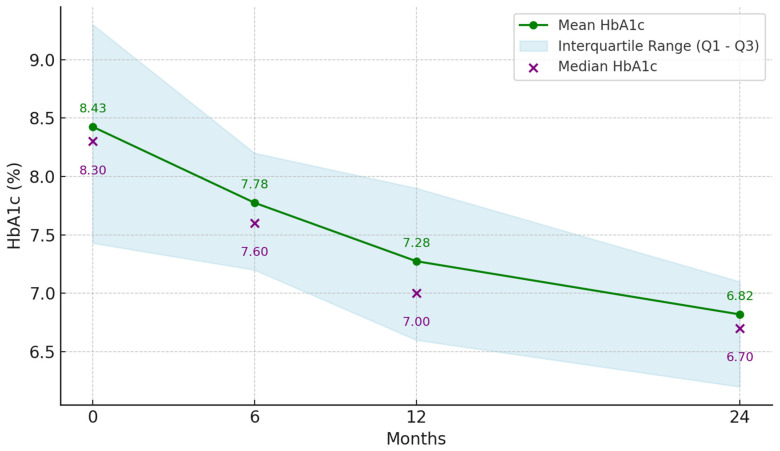
Mean glycated hemoglobin (HbA1c) at 6, 12, and 24 months after dulaglutide treatment initiation (n = 55).

**Figure 4 jcm-14-03536-f004:**
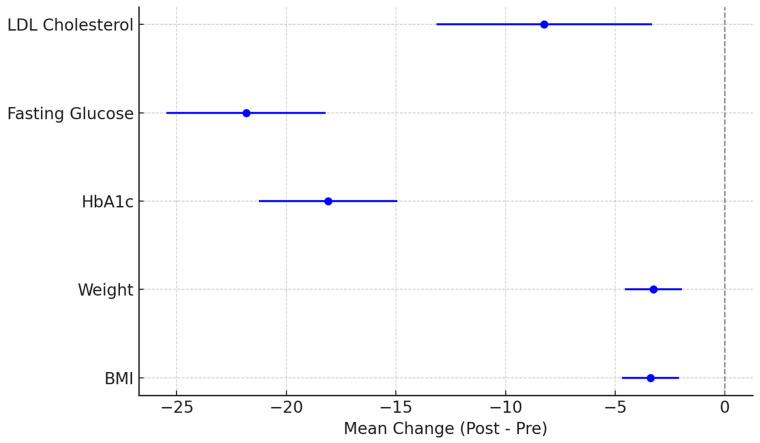
Clinical parameter changes at 24 months: mean differences and 95% confidence intervals.

**Table 1 jcm-14-03536-t001:** Baseline characteristics (n = 55).

Variables	N = 55
Age, years	58.2 (average 33–77) years
Men, n (%)	30 (54.5%)
Diabetes duration (years)	9.5 (3–19) years
Arterial hypertension	29 (52.7%)
Complications	
Diabetic peripheral polyneuropathy	15 (27.3%)
Diabetic retinopathy	9 (16.4%)
Diabetic nephropathy	10 (18.1%)
Type 2 diabetes treatment	
Metformin	13 (23.6%)
Metformin + Sulfonylureas	34 (61.9%)
Metformin + DPP − 4 inhibitors	5 (9.1%)
Sulfonylureas	3 (5.5%)
Non-insulin treated patients	48 (87.3%)

**Table 2 jcm-14-03536-t002:** Parameters at baseline, and at 6, 12, and 24 months after initiation of dulaglutide treatment (n = 55).

Parameter	Baseline	After 6 Months	After 12 Months	After 24 Months	*p*-Value
Fasting plasma glucose (mg/dL)	179 IQR = [152, 201]	161 IQR = [140, 180]	149.5 IQR = [135, 168]	136.4 IQR = [123, 149]	*p* < 0.001 *
Glycated hemoglobin (HbA1c) (%)	8.4 IQR = [7.43, 9.3]	7.7 IQR = [7.2, 8.2]	7.2 IQR = [6.6, 7.9]	6.8 IQR = [6.2, 7.1]	*p* < 0.001 *
Body weight (kg)	95.2 IQR = [83.5, 102]	-	-	91.8 IQR = [80, 100]	*p* < 0.001 *
Body mass index (kg/m^2^)	33.4 IQR = [28.6, 37.7]	-	-	32.2 IQR = [27.72, 35.79]	*p* < 0.001 *
LDL-cholesterol	112.9	-	-	99.2	*p* < 0.001 *

* The *p*-values < 0.001 are valid for the comparison of all parameters between baseline vs. 6 months, 6 months vs. 12 months, 12 months vs. 24 months, and baseline vs. 24 months.

**Table 3 jcm-14-03536-t003:** Paired samples statistical comparisons (n = 55).

Parameter Comparison	Test Used	Test Statistic	*p*-Value	Interpretation
BMI baseline vs. 24 months	*t*-test	t = 4.592	*p* < 0.001	Significant reduction in BMI
Body weight baseline vs. 24 months	Wilcoxon	Z = −4.450	*p* < 0.001	Significant reduction in weight
HbA1c baseline vs. 6 months	*t*-test	t = 7.422	*p* < 0.001	Significant improvement in HbA1c
HbA1c 6 months vs. 12 months	Wilcoxon	Z = −5.168	*p* < 0.001	Continued improvement in HbA1c
HbA1c 12 months vs. 24 months	Wilcoxon	Z = −5.108	*p* < 0.001	Sustained improvement in HbA1c
FPG baseline vs. 6 months	Wilcoxon	Z = −4.509	*p* < 0.001	Significant reduction in fasting glucose
FPG 6 months vs. 12 months	Wilcoxon	Z = −4.310	*p* < 0.001	Further decrease in fasting glucose
FPG 12 months vs. 24 months	Wilcoxon	Z = −4.974	*p* < 0.001	Continued improvement in fasting glucose
LDL baseline vs. 24 months	Wilcoxon	Z = −3.427	*p* < 0.001	Significant reduction in LDL-cholesterol

BMI = Body Mass Index; HbA1c = glycated hemoglobin; FPG = fasting plasma glucose; LDL-C = low-density lipoprotein cholesterol. *p* < 0.001 values indicate high statistical significance.

## Data Availability

The original contributions presented in this study are included in the article. Further inquiries can be directed to the corresponding author.

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
