# Peer review of "Dulaglutide 1.5 mg Significantly Improves Glycemic Control and Lowers LDL-Cholesterol and Body Weight in Romanian Patients with Type 2 Diabetes"

_jcm, 2025, doi:10.3390/jcm14103536_

Round 1
Reviewer 1 Report
Comments and Suggestions for Authors
The manuscript presents a retrospective observational study evaluating the impact of dulaglutide 1.5 mg on glycemic control, LDL-cholesterol, and weight over a 24-month period in Romanian patients with type 2 diabetes mellitus. The findings are broadly consistent with existing literature and are of interest to clinicians managing T2DM in resource-constrained or region-specific settings. However, there are several areas where the manuscript requires significant improvement:
The study reports significant weight and BMI reductions with dulaglutide therapy, but does not assess or even discuss the nature of this weight loss. This is a critical omission. Recent studies, including a comprehensive review in Metabolism(DOI: 10.1016/j.metabol.2024.156113), emphasize that GLP-1 RAs may lead to concurrent loss of lean muscle mass—a concern particularly in aging, multimorbid, or sarcopenia-prone populations. Are there data to perform this analysis? If not it should be discussed at least.
While the authors cite multiple AWARD studies, comparative context with real-world observational cohorts is limited. For example, several multicenter European registry data or pragmatic trials have evaluated dulaglutide effectiveness and could strengthen the relevance of the findings. Additionally, the clinical significance of a 3.3 kg weight loss or 1.2 kg/m² BMI reduction over 24 months is modest and should be interpreted with nuance.
The heterogeneity in background antihyperglycemic therapy (metformin alone, sulfonylureas, DPP-4 inhibitors, insulin) is acknowledged but insufficiently explored. The impact of these agents, particularly sulfonylureas and insulin, on weight and glycemia could confound the observed effects of dulaglutide. Perform or at least discuss subgroup comparisons (if sample size permits).
Author Response
Comment 1: The manuscript presents a retrospective observational study evaluating the impact of dulaglutide 1.5 mg on glycemic control, LDL-cholesterol, and weight over a 24-month period in Romanian patients with type 2 diabetes mellitus. The findings are broadly consistent with existing literature and are of interest to clinicians managing T2DM in resource-constrained or region-specific settings. However, there are several areas where the manuscript requires significant improvement
Response 1:
Thank you for this revision, which undoubtedly complements and enhances the scientific quality of the article, while further highlighting the value of the results obtained.
Comment 2: The study reports significant weight and BMI reductions with dulaglutide therapy, but does not assess or even discuss the nature of this weight loss. This is a critical omission. Recent studies, including a comprehensive review in Metabolism(DOI: 10.1016/j.metabol.2024.156113), emphasize that GLP-1 RAs may lead to concurrent loss of lean muscle mass—a concern particularly in aging, multimorbid, or sarcopenia-prone populations. Are there data to perform this analysis? If not it should be discussed at least.
Response 2:
We have added two substantial paragraphs at the end of the discussion sections, in which we reviewed the literature and commented on the effects of GLP-1 RA treatment on muscle loss, including the effects of dulaglutide. Unfortunately, we did not assess muscle mass loss in the patients from our study; nevertheless, your suggestion to address the potential sarcopenic effect of GLP-1 RAs is extremely valuable.
Comment 3:
While the authors cite multiple AWARD studies, comparative context with real-world observational cohorts is limited. For example, several multicenter European registry data or pragmatic trials have evaluated dulaglutide effectiveness and could strengthen the relevance of the findings. Additionally, the clinical significance of a 3.3 kg weight loss or 1.2 kg/m² BMI reduction over 24 months is modest and should be interpreted with nuance.
Response 3:
We have added several non-observational studies in which we compared the results, and indeed they were similar. Unfortunately, most of the available studies report their weight loss outcomes at 6 months, whereas in our study, we assessed weight reduction directly at 24 months. For this reason, the references cited in the text are limited.
Comment 4:
The heterogeneity in background antihyperglycemic therapy (metformin alone, sulfonylureas, DPP-4 inhibitors, insulin) is acknowledged but insufficiently explored. The impact of these agents, particularly sulfonylureas and insulin, on weight and glycemia could confound the observed effects of dulaglutide. Perform or at least discuss subgroup comparisons (if sample size permits).
Response 4:
Tthank you for the insightful observation regarding the heterogeneity of background antihyperglycemic therapy. We acknowledge that prior treatments, particularly sulfonylureas and insulin, may influence both glycemic control and body weight, which could act as confounding factors in interpreting the effects of dulaglutide. We considered the possibility of performing subgroup comparisons based on background therapy. Unfortunately, the relatively small sample size did not allow for robust statistical analyses. Nevertheless, we have added a note in the discussion section addressing the potential impact of background medication and the limitations this introduces in the interpretation of our results.
Reviewer 2 Report
Comments and Suggestions for Authors
The manuscript provides valuable real-world data in a previously underrepresented population (Romanian patients with T2DM), highlighting the efficacy of dulaglutide. The study demonstrates statistically significant improvements in HbA1c, fasting plasma glucose (FPG), LDL-cholesterol, and body weight after 24 months.
Some observations for this study contribute relevant regional data, and several methodological, statistical, and interpretative aspects need clarification or improvement to strengthen its scientific rigor.
All patients were on statins at baseline; LDL reduction may be partially attributable to statins rather than dulaglutide. Please add a discussion about the potential confounding effect of concurrent lipid-lowering therapy.
There is a typo error on gender differences; are reported non-significant differences between sexes (p>0.005), but the p-value threshold for significance should be p>0.05, is likely a typo.
Mean HbA1c drop from 8.4% to 6.8% is statistically significant, but fewer patients reached HbA1c <7% compared to AWARD trials (only 58% at 24 months). The clinical implication of not achieving the target in 42% of patients needs more discussion.
I recommend acknowledging polypharmacy and heterogeneity of treatment regimens as possible confounders as statins, sulfonylureas, and other antidiabetics. Also, adding data on adverse events (nausea, GI side effects of GLP-1RA), discontinuation rates, and hospitalizations Even in retrospective chart reviews, these data should be collected if they are available.
Author Response
Comment 1:
The manuscript provides valuable real-world data in a previously underrepresented population (Romanian patients with T2DM), highlighting the efficacy of dulaglutide. The study demonstrates statistically significant improvements in HbA1c, fasting plasma glucose (FPG), LDL-cholesterol, and body weight after 24 months.
Some observations for this study contribute relevant regional data, and several methodological, statistical, and interpretative aspects need clarification or improvement to strengthen its scientific rigor.
Response 1:
We thank the reviewer for the pertinent and valuable comments, which have indeed enhanced the scientific rigor of the manuscript.
Comment 2:
All patients were on statins at baseline; LDL reduction may be partially attributable to statins rather than dulaglutide. Please add a discussion about the potential confounding effect of concurrent lipid-lowering therapy.
Response 2:
We have added a paragraph in the discussion section addressing the effects of dulaglutide on LDL-cholesterol reduction, along with appropriate references. Thank you for the suggestion—indeed, this clarification was necessary and has now been included.
Comment 3:
There is a typo error on gender differences; are reported non-significant differences between sexes (p>0.005), but the p-value threshold for significance should be p>0.05, is likely a typo.
Response 3:
We have corrected it, thank you very much
Comment 4:
Mean HbA1c drop from 8.4% to 6.8% is statistically significant, but fewer patients reached HbA1c <7% compared to AWARD trials (only 58% at 24 months). The clinical implication of not achieving the target in 42% of patients needs more discussion.
Response 4:
In the discussion section on glycemic control, we added a substantial paragraph in which we explained the significantly smaller reductions in glycated hemoglobin among Romanian patients, and specifically discussed the particularities of the Romanian population
Comment 5
I recommend acknowledging polypharmacy and heterogeneity of treatment regimens as possible confounders as statins, sulfonylureas, and other antidiabetics. Also, adding data on adverse events (nausea, GI side effects of GLP-1RA), discontinuation rates, and hospitalizations Even in retrospective chart reviews, these data should be collected if they are available.
Response 5:
A comprehensive paragraph was added to the discussion section addressing polypharmacy, treatment regimen heterogeneity, and potential confounding factors. Regarding adverse events, unfortunately, systematic monitoring was not conducted; however, all 55 patients tolerated dulaglutide well.
Round 2
Reviewer 1 Report
Comments and Suggestions for Authors
While the limited sample size understandably restricted the authors from performing additional subgroup analyses that could have substantially strengthened the manuscript, it remains essential to more clearly acknowledge a key nuance in body composition outcomes. Although GLP-1 receptor agonists markedly reduce total and fat mass, their effect on relative lean mass appears negligible. As shown in a recent network meta-analysis (10.1016/j.metabol.2024.156113), approximately 25% of the overall weight loss with GLP-1RAs and dual GLP-1/GIP-RAs is attributable to lean mass reduction; however, the proportion of lean mass relative to total body weight remains stable. This finding suggests that while absolute lean mass declines, overall body composition is largely preserved—a distinction with important implications for clinical interpretation.
Author Response
Comment 1:
While the limited sample size understandably restricted the authors from performing additional subgroup analyses that could have substantially strengthened the manuscript, it remains essential to more clearly acknowledge a key nuance in body composition outcomes. Although GLP-1 receptor agonists markedly reduce total and fat mass, their effect on relative lean mass appears negligible. As shown in a recent network meta-analysis (10.1016/j.metabol.2024.156113), approximately 25% of the overall weight loss with GLP-1RAs and dual GLP-1/GIP-RAs is attributable to lean mass reduction; however, the proportion of lean mass relative to total body weight remains stable. This finding suggests that while absolute lean mass declines, overall body composition is largely preserved—a distinction with important implications for clinical interpretation.
Response 1:
We thank the reviewer for this insightful and constructive comment. As recommended, we have now incorporated the requested clarification regarding body composition outcomes into the revised manuscript. Specifically, we addressed the distinction between absolute and relative lean mass changes associated with GLP-1 receptor agonists and dual GLP-1/GIP receptor agonists. The updated paragraph highlights the preservation of overall body composition despite reductions in absolute lean mass, based on recent meta-analytic evidence (10.1016/j.metabol.2024.156113).
To facilitate review, the newly added text has been highlighted in blue within the manuscript. We appreciate the reviewer’s suggestion, which helped enhance the clarity and clinical relevance of our discussion.
Comment 1:
While the limited sample size understandably restricted the authors from performing additional subgroup analyses that could have substantially strengthened the manuscript, it remains essential to more clearly acknowledge a key nuance in body composition outcomes. Although GLP-1 receptor agonists markedly reduce total and fat mass, their effect on relative lean mass appears negligible. As shown in a recent network meta-analysis (10.1016/j.metabol.2024.156113), approximately 25% of the overall weight loss with GLP-1RAs and dual GLP-1/GIP-RAs is attributable to lean mass reduction; however, the proportion of lean mass relative to total body weight remains stable. This finding suggests that while absolute lean mass declines, overall body composition is largely preserved—a distinction with important implications for clinical interpretation.
Response 1:
We thank the reviewer for this insightful and constructive comment. As recommended, we have now incorporated the requested clarification regarding body composition outcomes into the revised manuscript. Specifically, we addressed the distinction between absolute and relative lean mass changes associated with GLP-1 receptor agonists and dual GLP-1/GIP receptor agonists. The updated paragraph highlights the preservation of overall body composition despite reductions in absolute lean mass, based on recent meta-analytic evidence (10.1016/j.metabol.2024.156113).
To facilitate review, the newly added text has been highlighted in blue within the manuscript. We appreciate the reviewer’s suggestion, which helped enhance the clarity and clinical relevance of our discussion.
Reviewer 2 Report
Comments and Suggestions for Authors
The manuscript now includes improved discussion, stratification of treatment regimens, and context that makes it more easy to follow. However, the lack of statistical clarity and the absence of adverse event data remain concerns that must be addressed before acceptance. Still only reports p-values; effect sizes and CIs are missing. No description of how normality was assessed to choose the t-test vs. non-parametric tests. Even in a retrospective study, authors should comment on adverse effects commonly associated with GLP-1RAs (nausea, vomiting, discontinuation).
The authors discussed well Eastern European glycemic control gaps and sociocultural barriers to optimal diabetes care.
Author Response
Comment 1: The manuscript now includes improved discussion, stratification of treatment regimens, and context that makes it more easy to follow. However, the lack of statistical clarity and the absence of adverse event data remain concerns that must be addressed before acceptance. Still only reports p-values; effect sizes and CIs are missing. No description of how normality was assessed to choose the t-test vs. non-parametric tests. Even in a retrospective study, authors should comment on adverse effects commonly associated with GLP-1RAs (nausea, vomiting, discontinuation).
Response 1:
We sincerely thank the reviewer for their careful reading and valuable suggestions, which have significantly improved the quality and clarity of our manuscript.
In response to the concerns raised, we have now incorporated additional statistical details, including effect sizes and confidence intervals where appropriate. These have been clearly presented in both the results section and newly formatted tables, which we hope will address the issue of statistical clarity and provide a more complete understanding of the outcomes. Additionally, we have explicitly described the method used to assess data normality (Shapiro–Wilk test) and clarified the rationale for selecting between parametric and non-parametric statistical tests.
As recommended, we have also expanded the manuscript to include a discussion of adverse effects commonly associated with GLP-1 receptor agonists, particularly dulaglutide.
We greatly appreciate the reviewer’s insightful feedback, which has helped us strengthen the manuscript both methodologically and clinically.
Comment 2:
The authors discussed well Eastern European glycemic control gaps and sociocultural barriers to optimal diabetes care.
Response 2:
We thank the reviewer for this kind remark. We are pleased that the discussion on regional disparities in glycemic control and the sociocultural challenges specific to Eastern Europe was found relevant and well-articulated. Our intention was to provide meaningful context for interpreting treatment outcomes in this population, and we appreciate the reviewer’s acknowledgement of this effort.
To facilitate review, the newly added text has been highlighted in blue within the manuscript. We appreciate the reviewer’s suggestion, which helped enhance the clarity and clinical relevance of our discussion.